# Transient amplifiers of selection and reducers of fixation for death-Birth updating on graphs

**Benjamin Allen**[1]*, **Christine Sample**[1], **Robert Jencks**[1], **James Withers**[1],
**Patricia Steinhagen**[1], **Lori Brizuela**[1], **Joshua Kolodny**[1], **Darren Parke**[1], **Gabor Lippner**[2],
**Yulia A. Dementieva**[1]

**1** Department of Mathematics, Emmanuel College, Boston, Massachusetts, United States of America,
**2** Department of Mathematics, Northeastern University, Boston, Massachusetts, United States of America

* allenb@emmanuel.edu

**Data Availability Statement:** All relevant data are within the manuscript and its Supporting Information files.

## Abstract

The spatial structure of an evolving population affects the balance of natural selection versus genetic drift. Some structures amplify selection, increasing the role that fitness differences play in determining which mutations become fixed. Other structures suppress selection, reducing the effect of fitness differences and increasing the role of random chance. This phenomenon can be modeled by representing spatial structure as a graph, with individuals occupying vertices. Births and deaths occur stochastically, according to a specified update rule. We study *death-Birth* updating: An individual is chosen to die and then its neighbors compete to reproduce into the vacant spot. Previous numerical experiments suggested that amplifiers of selection for this process are either rare or nonexistent. We introduce a perturbative method for this problem for weak selection regime, meaning that mutations have small fitness effects. We show that fixation probability under weak selection can be calculated in terms of the coalescence times of random walks. This result leads naturally to a new definition of effective population size. Using this and other methods, we uncover the first known examples of transient amplifiers of selection (graphs that amplify selection for a particular range of fitness values) for the death-Birth process. We also exhibit new families of "reducers of fixation", which decrease the fixation probability of all mutations, whether beneficial or deleterious.

## Author summary

Natural selection is often thought of as "survival of the fittest", but random chance plays a significant role in which mutations persist and which are eliminated. The balance of selection versus randomness is affected by spatial structure—how individuals are arranged within their habitat. Some structures amplify the effects of selection, so that only the fittest mutations are likely to persist. Others suppress the effects of selection, making the survival of genes primarily a matter of random chance. We study this question using a mathematical model called the "death-Birth process". Previous studies have found that spatial structure rarely, if ever, amplifies selection for this process. Here we report that spatial structure can indeed amplify selection, at least for mutations with small fitness

**Funding:** BA, CS, LB, RJ, JK, and PS were supported by the National Science Foundation Award #1715315 (https://nsf.gov/awardsearch/showAward?AWD_ID=1715315&HistoricalAwards=false). The funders had no role in study design, data collection and analysis, decision to publish, or preparation of the manuscript.

**Competing interests:** The authors have declared that no competing interests exist.

effects. We also identify structures that reduce the spread of any new mutation, whether beneficial or deleterious. Our work introduces new mathematical techniques for assessing how population structure affects natural selection.

## Introduction

Spatial population structure has a variety of effects on natural selection [1–5]. These effects can be studied mathematically by representing spatial structure as a graph [3]. The vertices represent individuals, and the edges indicate spatial relationships between them. This modeling approach, known as evolutionary graph theory, has illuminated the effects of spatial structure on the rate of genetic change [6], the balance of selection versus neutral drift [3, 7, 8], and the evolution of cooperation and other social behaviors [4, 5, 9–15].

Here we focus on how spatial structure affects fixation probability—the probability that a new mutation will spread throughout the population, depending on its effect on fitness. Previous work [3, 7, 8, 16–27] has shown that some graphs act as *amplifiers* of selection, increasing the fixation probability of beneficial mutations, while reducing that of deleterious mutations. Other graphs act as *suppressors* of selection, increasing the fixation probability of deleterious mutations and reducing that of beneficial mutations. Over time, a population that is structured as an amplifier will more rapidly accrue beneficial mutations, whereas one structured as a suppressor will experience greater effects of random drift.

To be precise, the terms amplifier and suppressor cannot be ascribed solely to a graph itself. Fixation probabilities also depend on the *update rule*: the scheme by which births and deaths are determined. The majority of works on amplifiers and suppressors use *Birth-death (Bd)* updating: An individual is selected to reproduce proportionally to fitness, and its offspring replaces a uniformly-chosen neighbor. A minority of works [18, 23, 28, 29] have considered *death-Birth (dB)* updating: A uniformly-chosen individual dies, and a neighbor is chosen proportionally to fitness to reproduce into the vacancy. (Following Hindersin and Traulsen [23], we use uppercase letters for a demographic step that is affected by fitness, and lowercase letters for a step that is fitness-independent). Interestingly, the choice of update rule has a marked effect on fixation probabilities. For example, the Star graph (Fig 1B) is an amplifier of selection for Bd updating [3, 17] (so long as the initial mutant vertex is chosen uniformly at random [21]), but a suppressor for dB updating [18].

A recent numerical investigation [23] of thousands of random graphs up to size 14 found no amplifiers of selection for death-Birth updating. This suggests that amplifiers for dB are either nonexistent or rare, at least among small graphs. This work also identified a graph (the cycle; Fig 1C) that, for dB updating, reduces fixation probabilities for *all* mutations that affect fitness, whether beneficial or deleterious. The cycle is therefore neither an amplifier nor a suppressor; it might instead be called a "reducer of fixation", in that it preserves the resident wild-type regardless of fitness effects. A follow-up work [30] identified other reducers of fixation.

Here we investigate fixation probabilities for death-Birth updating on graphs, using a variety of analytical and numerical methods. We develop a weak-selection approach to this question, based on coalescing random walk methods [31, 32] that were previously used to study evolutionary games on graphs [5, 10, 14]. Weak selection means that the fitness of the mutant is close to that of the resident; i.e., the mutation is either slightly beneficial or slightly deleterious. Unlike earlier numerical methods [23, 26, 33], the weak-selection method can be

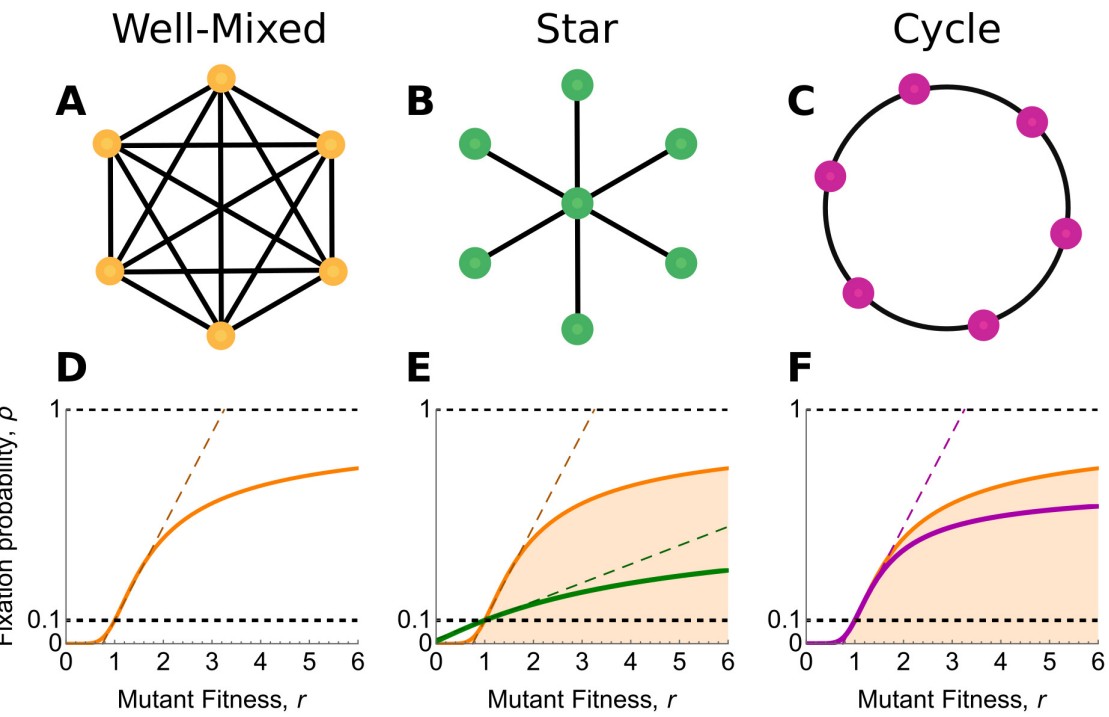

**Fig 1. Fixation probabilities for constant selection on graphs.** (**A**) The complete graph represents a well-mixed population. (**B**) The star graph consists of one hub vertex connected to $n$ leaf vertices. This star is a suppressor of selection for death-Birth updating [18]. (**C**) The cycle, a regular graph of degree 2, is a reducer of fixation: the fixation probability of any mutant type of fitness $r \neq 1$ is smaller than it would be in the well-mixed case [23]. Panels (**D**)–(**F**) plot fixation probability versus mutant fitness for the respective graphs, with the well-mixed case (orange curve) shown for comparison. Dashed lines show the linear approximation to fixation probability at $r = 1$. These approximations are accurate for weak selection ($r \approx 1$) and can be computed from coalescence times using Eqs (5)–(9).

performed in polynomial time, allowing for efficient identification of amplifiers and suppressors of weak selection. We apply this method to several graph families and random graph models. We also compute fixation probabilities for arbitrary mutant fitness (beyond weak selection) for these graph families.

We find, contrary to the expectation set by previous numerical experiments [23], that amplifiers, of a sort, do exist for death-Birth updating. Specifically, we exhibit several families of *transient amplifiers*, which amplify selection only for a certain range of mutant fitness values. We also uncover new examples of reducers of fixation.

Our weak-selection method also leads to new theoretical results. First, the form of our expression for fixation probability suggests a new definition of effective population size, with intriguing connections to previous definitions [16, 34–40]. Second, we show that for *isothermal* graphs—which have the same edge weight sum at each vertex—the fixation probability coincides, under weak selection, with that of a well-mixed population. This result is reminiscent of the Isothermal Theorem of Lieberman et al. [3], which applies to Bd updating (see also Refs. [15, 28, 29]). However, whereas the original Isothermal Theorem is valid for any strength of selection, our new result applies only to weak selection. Third, we exhibit a recipe by which amplifiers of weak selection can be constructed as perturbations of isothermal graphs. Finally, we show that fixation probabilities under weak selection can be well-approximated using only the first two moments of the degree distribution. This approximation helps explain why amplifiers of selection (even transient ones) are rare for dB updating.

## Methods

### Model

We study an established model of natural selection on graphs [3, 7, 8, 16–29, 41–43]. Spatial structure is represented as a connected, weighted, undirected graph $G$. Joining each pair of vertices $i$ and $j$ is an edge of weight $w_{ij} \geq 0$, with $w_{ij} = w_{ji}$ since $G$ is undirected. We exclude the possibility of self-loops by setting $w_{ii} = 0$ for each vertex $i$. The size of the graph, which is also the population size, is denoted $N$.

Each vertex houses a single haploid individual. Individuals can be of mutant or resident (wild-) type. Mutants have fitness $r > 0$, while the fitness of the resident type is set to 1. Advantageous mutants have $r > 1$, while deleterious mutants have $r < 1$. The case $r = 1$ describes neutral drift, for which the mutation has no fitness effect. This model describes *constant selection*, in that the fitnesses of the competing types do not vary with the current population state.

Selection proceeds according to the death-Birth (dB) update rule [4, 18, 44]. First, an individual is selected uniformly at random for death, creating a vacant vertex. Then, a neighbor of the vacant vertex is chosen to reproduce, with probability proportional to (fitness) × (edge weight to the vacant vertex). The new offspring fills the vacancy, inheriting the type of the parent.

As an initial state, we suppose that a single mutant is introduced, at a vertex chosen uniformly at random, in a population otherwise composed of residents. We define the mutation's *fixation probability* as the expected probability that a state of all mutants is reached from this initial condition. The fixation probability of a mutation of fitness $r$ on a graph $G$ is denoted $\rho_G(r)$.

The baseline case of a well-mixed population is represented by the complete graph $K_N$ of size $N$ (Fig 1A). For dB updating on the complete graph $K_N$, a mutant of fitness $r$ has fixation probability [23, 28]

$$\rho_{K_N}(r) = \frac{N-1}{N} \frac{1 - r^{-1}}{1 - r^{-(N-1)}}. \tag{1}$$

We characterize the effects of graph structure on fixation probabilities using the following definitions:

**Definition** Let $G$ be a graph of size $N$. Then $G$ is

- An *amplifier of selection* if $\rho_G(r) < \rho_{K_N}(r)$ for $0 < r < 1$ and $\rho_G(r) > \rho_{K_N}(r)$ for $r > 1$.

- A *suppressor of selection* if $\rho_G(r) > \rho_{K_N}(r)$ for $0 < r < 1$ and $\rho_G(r) < \rho_{K_N}(r)$ for $r > 1$.

- A *transient amplifier of selection* if there is some $r^* > 1$ such that $\rho_G(r) < \rho_{K_N}(r)$ for $0 < r < 1$ and for $r > r^*$, and $\rho_G(r) > \rho_{K_N}(r)$ for $1 < r < r^*$.

- A *reducer of fixation* if $\rho_G(r) < \rho_{K_N}(r)$ for all $r \neq 1$.

For example, the star graph $S_n$ with $n$ leaves (population size $N = n + 1$; Fig 1B) is a suppressor of selection for dB updating [18], with fixation probability [45]

$$\rho_{S_n}(r) = \frac{(N-1)r + 1}{N(r+1)} \left( \frac{1}{N} + \frac{r}{N + 2r - 2} \right). \tag{2}$$

The cycle $C_N$ is a reducer of fixation for dB updating [23], with fixation probability [28]

$$\rho_{C_N}(r) = \frac{2(r-1)}{3r - 1 + r^{-(N-3)} - 3r^{-(N-2)}}. \tag{3}$$

Other examples of reducers were identified by Hindersin et al. [30], who called them "suppressors of evolution"; we prefer "reducers of fixation" to avoid confusion with suppressors of selection.

A companion work [46] proves that there are no (non-transient) amplifiers of selection for dB updating. Transient amplifiers of selection were previously known for Bd updating [19] but not for dB updating. For Bd updating, there are some graphs that do not fit any of the above definitions, but alternate between amplification and suppression (i.e, $\rho_G(r) > \rho_{K_N}(r)$ on a disconnected set of $r$-values) [27]; such examples have not been discovered for dB updating.

## Analysis of weak selection

Fixation probabilities on graphs can be difficult to compute. Current numerical methods [22, 23, 26, 33] involve solving a system of $\mathcal{O}(2^N)$ equations to compute fixation probabilities on a given graph of size $N$. For this reason, previous analyses have focused on graphs that are small [23, 26, 27, 33, 42, 43], highly symmetric [3, 7, 17, 19–21, 24, 25], or are constrained in the types of connections between vertices [47].

One way to mitigate these difficulties is to focus on *weak selection*, which is the regime $r \approx 1$. Weak selection can be studied as a perturbation of neutral drift ($r = 1$). This approach has been fruitfully applied to population genetics [48–50] and evolutionary game theory [4, 5, 10, 11, 14, 44, 51], but so far has not been applied to models of constant selection on graphs.

To implement weak selection for our model, we write the fitness of the mutant as $r = 1 + \delta$, with $\delta$ representing the mutation's selection coefficient. We consider the first-order Taylor expansion of the fixation probability, $\rho_G(1 + \delta)$, at $\delta = 0$. For the complete graph, Taylor expansion of Eq (1) yields

$$\rho_{K_N}(1 + \delta) = \frac{1}{N} + \delta \frac{N - 2}{2N} + \mathcal{O}(\delta^2). \tag{4}$$

## Coalescing random walks

For an arbitrary weighted, connected graph, we apply a method developed by Allen et al. [5] to calculate fixation probabilities under weak selection. This method uses *coalescing random walks*, which trace the co-ancestry of given individuals backwards in time to their most recent common ancestor.

Each individual's ancestry is represented as a random walk on $G$. These random walks are defined by the step probabilities $p_{ij} = w_{ij}/w_i$, where $w_i = \sum_{j \in G} w_{ij}$ is the *weighted degree* of vertex $i$. Importantly, $p_{ij}$ is also equal to the conditional probability, under neutral drift ($r = 1$), that $j$ reproduces, given that $i$ is replaced. Random walks on $G$ have a stationary distribution, in which the probability of vertex $i$ is equal to its relative weighted degree, $\pi_i = w_i/(\sum_{j \in G} w_j)$.

To represent the co-ancestry of two individuals, we consider a pair of random walkers. At each time-step, one of the two walkers is chosen (with equal probability) to take a step. The point at which the two walkers meet (coalesce) represents the most recent common ancestor. We let $\tau_{ij}$ denote the expected time to coalescence from initial vertices $i$ and $j$. These coalescence times can be determined from the following system of equations [5, 52]:

$$\tau_{ij} = \begin{cases} 0 & i = j \\ 1 + \frac{1}{2} \sum_{k \in G} \left( p_{ik}\tau_{jk} + p_{jk}\tau_{ik} \right) & i \neq j. \end{cases} \tag{5}$$

We also define the *remeeting time* $\tau_i$ from vertex $i$ as the expected time for two random walkers from vertex $i$ to rejoin each other. Remeeting times are related to coalescence times by

$$\tau_i = 1 + \sum_{j \in G} p_{ij} \tau_{ij}, \tag{6}$$

and obey the identity [5]

$$\sum_{i \in G} \pi_i^2 \tau_i = 1, \tag{7}$$

which is an instance of Kac's return time formula [53].

## Results

### Fixation probability under weak selection

Applying the properties of coalescence times, we prove in S1 Appendix that the fixation probability on an arbitrary (weighted, undirected, connected) graph $G$ can be expanded under weak selection as

$$\rho_G(1 + \delta) = \frac{1}{N} + \delta \frac{N_{\text{eff}} - 2}{2N} + \mathcal{O}(\delta^2), \tag{8}$$

where $N_{\text{eff}}$ is the *effective population size* of $G$, which we define as

$$N_{\text{eff}} = \sum_{i \in G} \pi_i \tau_i. \tag{9}$$

This definition of effective population size is distinct from, but closely related to, previous definitions [16, 34–40], as we review in the Discussion.

Comparing the first-order terms in Eqs (8) and (4) provides a criterion for the effects of graph structure on fixation probabilities under weak selection:

**Definition** Let $G$ be a graph of size $N$. We say $G$ is

- An *amplifier of weak selection* if $N_{\text{eff}} > N$,

- A *suppressor of weak selection* if $N_{\text{eff}} < N$.

An amplifier (respectively, suppressor) of weak selection is guaranteed to amplify (respectively, suppress) selection for all $r$ sufficiently close to 1. Formally, if $G$ is an amplifier of weak selection, there exist $a$, $b$ with $0 \leq a < 1 < b \leq \infty$ such that $\rho_G(r) < \rho_{K_N}(r)$ for $a < r < 1$ and $\rho_G(r) > \rho_{K_N}(r)$ for $1 < r < b$. Likewise, if $G$ is a suppressor of weak selection, there exist $a$, $b$ with $0 \leq a < 1 < b \leq \infty$ such that $\rho_G(r) > \rho_{K_N}(r)$ for $a < r < 1$ and $\rho_G(r) < \rho_{K_N}(r)$ for $1 < r < b$.

As an example, solving Eq (5) for the star graph $S_n$, and applying Eqs (6) and (9), we obtain $\tau_H = \tau_L = N_{\text{eff}} = 4n/(n + 1)$. Since the star graph has size $N = n + 1$, we find that the star is a suppressor of weak selection for all $n \geq 2$. Substituting in Eq (8), we obtain

$$\rho_G(1 + \delta) = \frac{1}{N} + \delta \frac{N - 2}{N^2} + \mathcal{O}(\delta^2), \tag{10}$$

which agrees with the Taylor expansion of Eq (2).

## Weak-selection Isothermal Theorem

A particularly interesting result arises in the special case of *isothermal* graphs. An undirected graph $G$ is isothermal if each vertex has the same weighted degree $w_i$, or equivalently, if $\pi_i = 1/N$ for each $i \in G$. The *Isothermal Theorem* [3] states that, for Bd updating, an isothermal graph has the same fixation probabilities as a well-mixed population of the same size, for all values of $r$ and all starting configurations of mutants. However, the corresponding statement for dB updating is false [28, 29]. For example, the cycle (Fig 1C) is isothermal, but its fixation probabilities, as given by Eq (3), differ from those of a well-mixed population, given by Eq (1).

Here we show that a weak-selection version of the isothermal theorem holds for death-Birth updating. For an isothermal graph $G$, Eqs (7) and (9) give

$$N_{\text{eff}} = \sum_{i \in G} \left(\frac{1}{N}\right) \tau_i = N \sum_{i \in G} \left(\frac{1}{N^2}\right) \tau_i = N \sum_{i \in G} \pi_i^2 \tau_i = N. \tag{11}$$

Combining with Eq (8), we arrive at the following result:

**Theorem** (Weak-Selection Isothermal Theorem for dB Updating). *Let G be a weighted, undirected, connected isothermal graph of size $N \geq 2$ with no self-loops. Then for dB updating, fixation probabilities on G coincide with those on the complete graph $K_N$ to first order in the selection coefficient $\delta$:*

$$\rho_G(1 + \delta) = \rho_{K_N}(1 + \delta) + \mathcal{O}(\delta^2). \tag{12}$$

In other words, if $G$ is isothermal, then the plots of $\rho_G(r)$ and $\rho_{K_N}(r)$ are tangent at $r = 1$. This implies that, for dB updating, isothermal graphs neither amplify nor suppress weak selection. For example, the cycle $C_N$ (Fig 1C) is isothermal, and therefore the plots of $\rho_{C_N}(r)$ and $\rho_{K_N}(r)$ are tangent at $r = 1$ (Fig 1F). However, these fixation probabilities do not coincide beyond $r = 1$; instead, $\rho_{C_N}(r) < \rho_{K_N}(r)$ for all $r \neq 1$ [33], meaning that the cycle is a reducer of fixation.

## Generating amplifiers of weak selection

The Weak-Selection Isothermal Theorem also suggests a method to generate amplifiers of weak selection via perturbations of an isothermal graph. Since $N_{\text{eff}} = N$ for all isothermal graphs, any perturbation that increases $N_{\text{eff}}$ will yield an amplifier of weak selection.

Consider a family of weighted graphs indexed by a parameter $\epsilon$, such that the graph is isothermal when $\epsilon = 0$. In S1 Appendix we derive the relationship

$$\left.\frac{dN_{\text{eff}}}{d\epsilon}\right|_{\epsilon=0} = -\sum_{i \in G} \left(\tau_i \frac{d\pi_i}{d\epsilon}\right)\bigg|_{\epsilon=0}. \tag{13}$$

Eq (13) suggests that we can construct amplifiers of weak selection by starting with an isothermal graph, and perturbing so as to decrease the relative weighted degree of vertices with large remeeting time and/or increase the relative weighted degree of vertices with small remeeting time.

Fig 2 provides an example to illustrate this method. Starting with an unweighted 3-regular graph of size 12, we reduce the weight of a single edge that is adjacent to a vertex of large remeeting time. This creates a family of amplifiers of weak selection for dB. Notably, the graph remains an amplifier even when this edge is deleted completely.

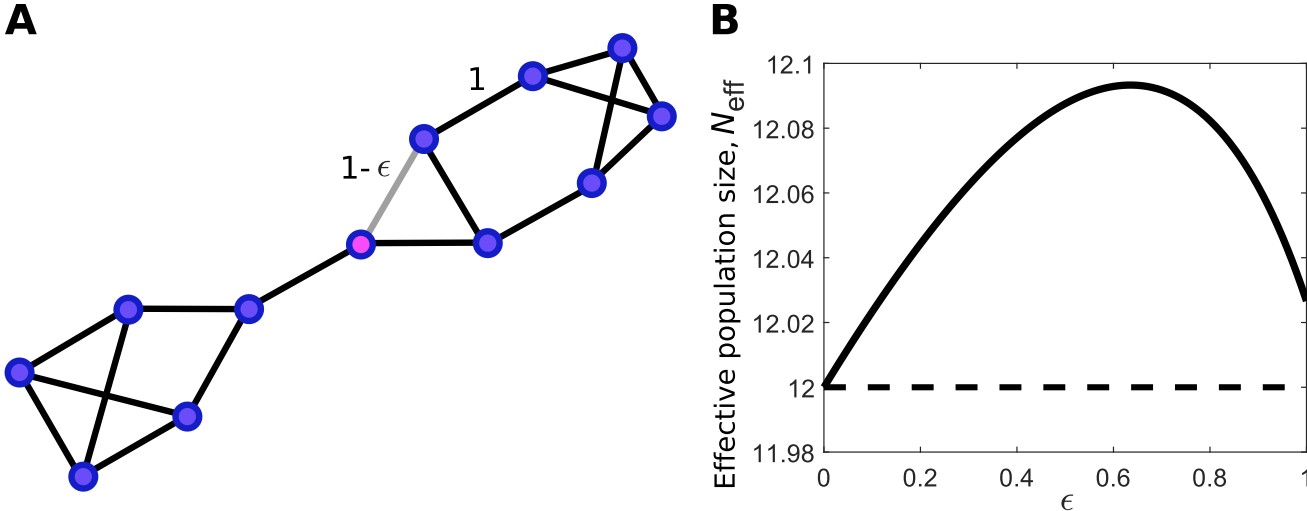

**Fig 2. Creating an amplifier of weak selection for death-Birth updating. (A)** We begin with a 3-regular graph of size 12 in which all edges have weight 1. This graph is isothermal, and therefore has $N_{\text{eff}} = N = 12$. We solve for remeeting times according to Eqs (5) and (6), and identify the vertex with the largest remeeting time ($\tau_i \approx 18.29$, shown in magenta). We decrease the edge weight from this vertex to one of its neighbors by an amount $\epsilon$. **(B)** As this edge weight decreases, the graph becomes an amplifier of weak selection ($N_{\text{eff}} > N$). For $\epsilon = 1$, the resulting undirected, unweighted graph is still an amplifier of weak selection, with effective population size $N_{\text{eff}} \approx 12.03$.

## Examples

We now introduce three example families of graphs, which can behave as transient amplifiers, suppressors, or reducers, depending on the parameter values. We analyze these graphs both for weak and nonweak selection. Our results are summarized in Table 1. Derivations and proofs are presented in S1 Appendix. Our analytical results are verified by Monte Carlo simulation in S1 and S2 Figs.

**Fan.**   The Fan, $F_{n,m}$, (Fig 3) has one hub and $n \geq 2$ blades. Each blade contains $m \geq 2$ vertices, for a total of $N = nm + 1$ vertices. Each blade vertex is joined to the hub by an edge of weight $\epsilon > 0$, and is joined to each other vertex on the same blade by an edge of weight 1. The Fan is isothermal when $\epsilon = (m-1)/(nm-1)$.

Applying our weak-selection method, we find that the Fan has effective population size

$$N_{\text{eff}} = N$$
$$+ \frac{(m-1-\epsilon(nm-1))(m(m-1)(n-2)+\epsilon(nm^2+nm-4m+2)+2\epsilon^2(nm-1))}{(m-1+2\epsilon)(m(m-1)+\epsilon(nm+2m-1)+\epsilon^2(nm+1))}. \quad (14)$$

**Table 1. Results for example graphs.**

| Example | Case | Classification |
|---|---|---|
| Separated Hubs *<br>($\epsilon \to 0$) | $n \leq h$ | Suppressor |
|  | $n = h + 1$ | Reducer |
|  | $n \geq h + 2$ | Transient amplifier |
| Star of Islands<br>($\epsilon \to 0$) | $m \leq h-1$ | Suppressor ** |
|  | $m = h$ | Reducer |
|  | $m \geq h + 1$ | Transient amplifier ** |

* The Fan is the $h = 1$ case of separated hubs.

** Proven only for weak selection (other cases are proven for arbitrary selection strength).

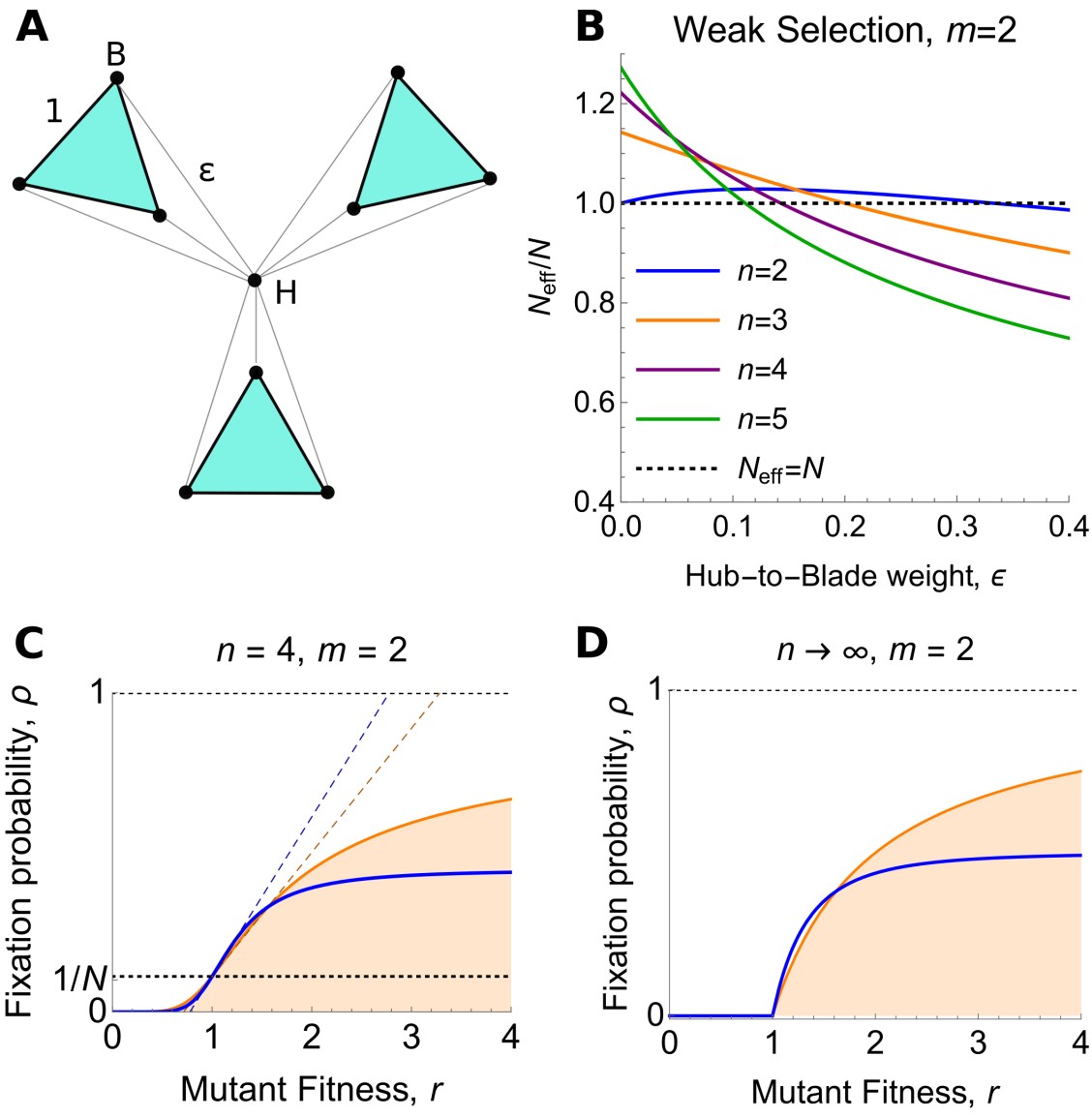

**Fig 3. The Fan.** (**A**) The Fan, $F_{n,m}$, consists of one hub and $n \geq 2$ "blades", with $m \geq 2$ vertices per blade. Edge weights are as shown. The case $n = m = 3$ is pictured. (**B**) The ratio of effective versus actual population size, plotted against the hub-to-blade edge weight $\epsilon$, for $m = 2$ vertices per blade. For $n = 2$ blades, the Fan is an amplifier of weak selection for $0 < \epsilon < 1/3$, but becomes a reducer in the $\epsilon \to 0$ limit. For $n \geq 3$, the Fan is a transient amplifier for sufficiently small $\epsilon$, including the $\epsilon \to 0$ limit. (**C**) Fixation probability for $F_{4,2}$ (blue curve), plotted against mutant fitness $r$, in the $\epsilon \to 0$ limit, according to Eq (15). The orange curve shows the corresponding well-mixed population result, Eq (1), for comparison. Dotted lines show the corresponding weak-selection results (i.e. the linear approximation at $r = 1$), according to Eqs (4), (8), and (14). (**D**) In the $n \to \infty$ limit, fixation probability is given by Eq (15), and the Fan is an amplifier for $1 < r < (1 + \sqrt{5})/2$.

From the sign of the second term, we observe that the Fan amplifies weak selection for all $0 < \epsilon < (m-1)/(nm-1)$ (Fig 3B).

Taking $\epsilon \to 0$, we obtain $N_{\mathrm{eff}} = nm + n - 1$. Although fixation is impossible when $\epsilon$ is exactly zero (because the population is disconnected in this case), the $\epsilon \to 0$ limit is still well-defined in the sense that $N_{\mathrm{eff}}$ can be made arbitrarily close to $nm + n - 1$ by choosing $\epsilon$ sufficiently small. In this limit, the Fan amplifies weak selection ($N_{\mathrm{eff}} > N$) for $n \geq 3$ blades, but neither amplifies nor suppresses weak selection ($N_{\mathrm{eff}} = N$) for $n = 2$. The strongest amplifier

of weak selection (largest $N_{\text{eff}}/N$) occurs for $m = 2$ and first $\epsilon \to 0$ and then $n \to \infty$; in this case, $N_{\text{eff}}/N \to 3/2$.

Moving beyond weak selection, we calculate the fixation probability for a mutation of arbitrary fitness $r > 0$, in the $\epsilon \to 0$ limit:

$$\rho_{F_{n,m}}(r) = \frac{n(m-1)(1-r^{-1})(1-r^{-(m+1)})}{(mn+1)(1-r^{-(m-1)})(1-r^{-n(m+1)})}. \tag{15}$$

In S2 Fig, we show excellent agreement between Eq (15) and Monte Carlo simulations for $\epsilon = 10^{-3}$. We prove in S1 Appendix that, in the $\epsilon \to 0$ limit, the Fan is a reducer of fixation for $n = 2$ and a transient amplifier of selection for all $n \geq 3$.

**Separated Hubs.**   Our next examples generalize the Fan graph in two different ways. First, we suppose that there are multiple hub vertices, which are not connected to each other. The resulting graph, which we call the *Separated Hubs* graph, $SH_{n,m,h}$ (Fig 4), has $h \geq 1$ hub vertices, $n \geq 2$ blades, and $m \geq 2$ vertices per blade for a total population size of $N = nm + h$. Vertices on the same blade are connected by edges of weight 1, and each blade vertex is connected to each hub by an edge of weight $\epsilon$. No other edges are present. The Fan is the $h = 1$ case of Separated Hubs.

The weak-selection results for arbitrary $\epsilon$ are rather cumbersome, but in the $\epsilon \to 0$ limit they simplify to

$$N_{\text{eff}} = nm + n - 1. \tag{16}$$

Interestingly, in this limit, the effective population size is independent of the number $h$ of hubs. Comparing Eq (16) to $N = nm + h$, we observe that the Separated Hubs graph (in the $\epsilon \to 0$ limit) is a suppressor of weak selection for $n \leq h$ and an amplifier of weak selection for $n \geq h + 2$. As for the Fan, the strongest amplifier of weak selection occurs for $m = 2$ and first $\epsilon \to 0$ and then $n \to \infty$, leading to $N_{\text{eff}}/N \to 3/2$. The strongest suppressor of weak selection (smallest $N_{\text{eff}}/N$) occurs for first $\epsilon \to 0$ and then $h \to \infty$, leading to $N_{\text{eff}}/N \to 0$.

Beyond weak selection, we compute the fixation probability for arbitrary $r > 0$ in the limit $\epsilon \to 0$:

$$\rho_{SH_{n,m,h}}(r) = \frac{n(m-1)(1-r^{-1})(1-r^{-(m+1)})}{(mn+h)(1-r^{-(m-1)})(1-r^{-n(m+1)})}. \tag{17}$$

In the limit of many blades, we obtain

$$\lim_{n\to\infty} \rho_{SH_{n,m,h}}(r) = \begin{cases} 0 & 0 \leq r \leq 1 \\ \dfrac{m-1}{m}\dfrac{(1-r^{-1})(1-r^{-(m+1)})}{1-r^{-(m-1)}} & r > 1. \end{cases} \tag{18}$$

We prove in S1 Appendix that the Separated Hubs graph, in the $\epsilon \to 0$ limit, is a suppressor for $n \leq h$, a transient amplifier for $n \geq h + 2$, and a reducer for $n = h + 1$.

**Star of Islands.**   Our final example, the *Star of Islands*, $SI_{n,m,h}$ (Fig 5), is obtained by joining the hubs in the Separated Hubs graph. It consists of $h \geq 2$ hub vertices and $n \geq 2$ islands, with $m \geq 2$ vertices per island, so that the total population size is again $N = nm + h$. Within the hub and within each island, vertices are connected to one another with weight 1. Additionally, each hub vertex is connected to each island vertex with weight $\epsilon > 0$.

For weak selection, in the $\epsilon \to 0$ limit, we calculate

$$N_{\text{eff}} = N + \frac{(m-h)mnh(h(h-1)+m(m-1)(n-2))}{(h(h-1)+m(m-1))(h(h-1)+m(m-1)n)}. \tag{19}$$

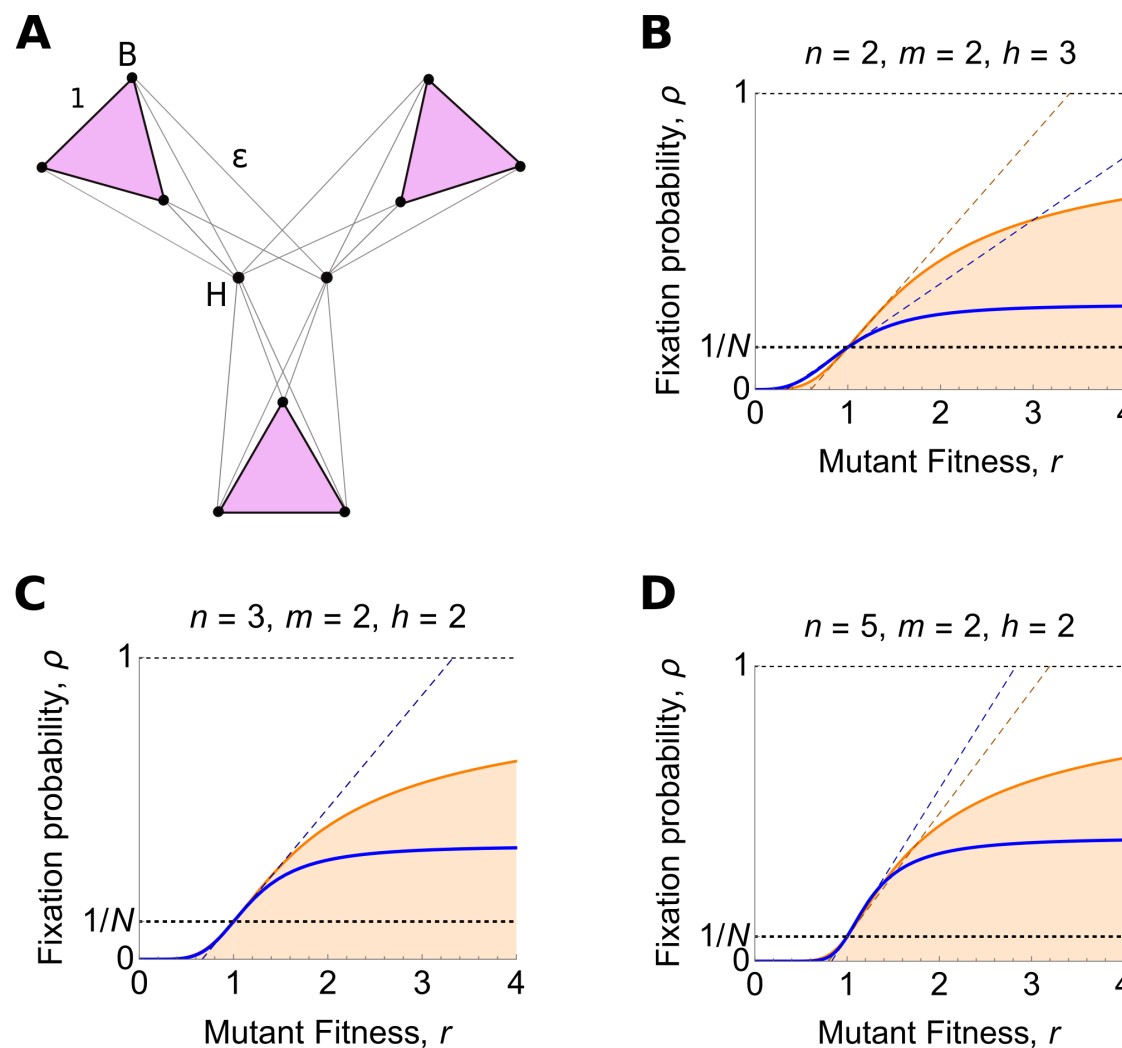

**Fig 4. Separated Hubs.** (A) The Separated Hubs graph consists of $h \geq 1$ hubs and $n \geq 2$ blades, with $m \geq 2$ vertices per blade. Edge weights are as shown. (B)–(D) Blue curves show fixation probability, Eq (17), plotted against mutant fitness $r$, in the $\epsilon \to 0$ limit. Blue dotted lines show the weak selection result, Eqs (8) and (16). The orange curve and dotted line show the corresponding well-mixed population results, Eqs (1) and (4), for comparison. The Separated Hubs graph is (B) a suppressor for $n \leq h$, (C) a reducer for $n = h + 1$, and (D) a transient amplifier for $n \geq h + 2$.

The second term on the right-hand side has the sign of $m - h$. It follows that the Star of Islands is an amplifier of weak selection when $m > h$, and a suppressor of weak selection when $m < h$.

We show in S1 Appendix that the strongest amplifier of weak selection occurs for $h = 2$, $m = 4$, and first $\epsilon \to 0$ and then $n \to \infty$. In this case $N_{\text{eff}}/N \to 9/7$. The strongest suppressor occurs for first $\epsilon \to 0$, then $n \to \infty$, and then $h \to \infty$, leading to $N_{\text{eff}}/N \to 0$.

For arbitrary $r > 0$, in the $\epsilon \to 0$ limit, we obtain $\rho_{SI_{n,m,h}}(r) = \text{num}/\text{denom}$ with

$$
\begin{aligned}
\text{num} = r^m(1 - r^{-1})(1 - r^{-(h+m)}) \\
(hr^h(1 - r^{-(h-1)})(mn(m-1)r^m + h(h-1)) \\
+ mr^m(1 - r^{-(m-1)})(mn(m-1) + h(h-1)r^h)),
\end{aligned}
\tag{20}
$$

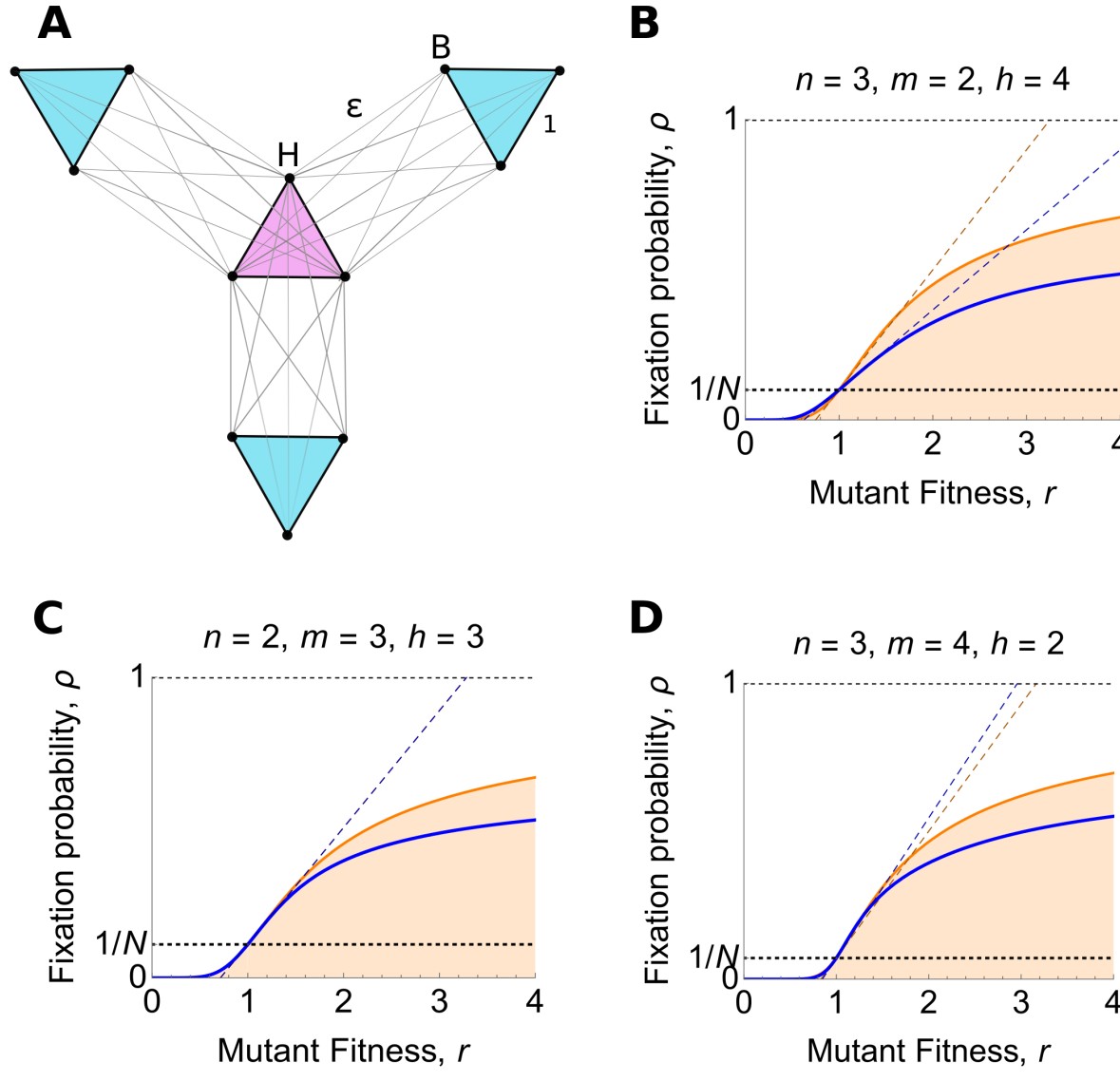

**Fig 5. Star of Islands.** (A) The Star of Islands graph consists of a hub island of size $h \geq 2$, and $n \geq 2$ other islands of size $m \geq 2$. Edge weights are as shown. (**B**)–(**D**) Blue curves show fixation probability, Eqs (20)–(22), plotted against mutant fitness $r$, in the $\epsilon \to 0$ limit. Blue dotted lines show the weak selection result, Eqs (8) and (19). The orange curve and dotted line show the corresponding well-mixed population results, Eqs (1) and (4), for comparison. The Star of Islands graph is (B) a suppressor for $m \leq h - 1$, (C) a reducer for $m = h$, and (D) a transient amplifier for $m \geq h + 1$.

$$\begin{aligned}
\text{denom} = (mn + h)(&h(1 - r^{-(h-1)}) \\
&+ mr^m(1 - r^{-(m-1)}))(mr^m(1 - r^{-(m-1)})(1 - x^n) \\
&+ h(1 - r^{-(h-1)})(r^{h+m} - x^n)),
\end{aligned} \quad (21)$$

and

$$x = \frac{mr^{-m}(r^{m-1} - 1) + h(r^{h-1} - 1)}{mr^h(r^{m-1} - 1) + h(r^{h-1} - 1)}. \quad (22)$$

In the limit of many islands, we obtain

$$\lim_{n\to\infty} \rho_{SI_{n,m,h}}(r) = \begin{cases} 0 & 0 \le r \le 1 \\ \dfrac{(m-1)(1-r^{-1})(1-r^{-(h+m)})}{hr^{-m}(1-r^{-(h-1)})+m(1-r^{-(m-1)})} & r > 1. \end{cases} \tag{23}$$

We prove in S1 Appendix that the Star of Islands is a reducer for $m = h$.

## Approximating fixation probability

We have defined the effective population size $N_{\text{eff}}$ in terms of the expected remeeting times of random walks. While this definition allows $N_{\text{eff}}$—and, via Eq (8), fixation probabilities under weak selection—to be computed in polynomial time, it gives little intuition for how $N_{\text{eff}}$ relates to more familiar graph statistics.

To build such intuition, we use a mean-field approximation from Fotouhi et al. [54]. We suppose that each remeeting time $\tau_i$ is approximately equal to a single value, $\tau$. Then from Eq (7) we have

$$1 = \sum_{i\in G} \pi_i^2 \tau_i \approx \tau \sum_{i\in G} \pi_i^2 = \frac{\tau\sum_{i\in G} w_i^2}{\left(\sum_{i\in G} w_i\right)^2} = \frac{\tau\mu_2}{N\mu_1^2}.$$

Above, $\mu_1 = \frac{1}{N}\sum_{i\in G} w_i$ and $\mu_2 = \frac{1}{N}\sum_{i\in G} w_i^2$ are the first and second moments, respectively, of the weighted degree distribution. Solving for $\tau$ and substituting in the definition of $N_{\text{eff}}$ gives the approximation

$$N_{\text{eff}} \approx N\mu_1^2/\mu_2. \tag{24}$$

Substituting in Eq (8) gives an approximation for fixation probability under weak selection in terms of $\mu_1$ and $\mu_2$. Interestingly, the right-hand side of Eq (24) was taken as the definition of effective population size by Antal et al. [16], who studied the same model but arrived at this expression by different methods and assumptions.

The approximation in Eq (24) is reasonably accurate when compared to exact numerical calculation of $N_{\text{eff}}/N$ for Erdös-Renyi and Barabási-Albert graphs (Fig 6). In particular, the approximation explains the general trend that larger, sparser, and more heterogeneous graphs act as stronger suppressors (have smaller $N_{\text{eff}}/N$ ratio). We note, however, that since $\mu_2 \le \mu_1^2$ for any degree distribution, the approximated $N_{\text{eff}}$ in Eq (24) is at most equal to the actual population size $N$, with equality only for isothermal graphs. Therefore, amplifiers of weak selection cannot be detected using this approximation.

## Discussion

### Weak-selection methodology

We have brought the method of weak selection, previously developed to analyze games on graphs [4, 5, 9–12, 14, 44], to bear on the question of amplifiers and suppressors. While our focus is on death-Birth updating, the method also applies to Birth-death updating, using a modified version of the coalescing random walk [5, 52]. Our weak-selection method has the advantage of being computable in polynomial time (in the size of the graph), in contrast to other numerical methods [23, 26, 27, 33], which take exponential time. Our expression for fixation probabilities in terms of coalescence times, Eq (8), also enables the proof of general results such as the Weak-Selection Isothermal Theorem for dB. A drawback of the weak-selection approach is that it does not distinguish between transient and non-transient amplifiers,

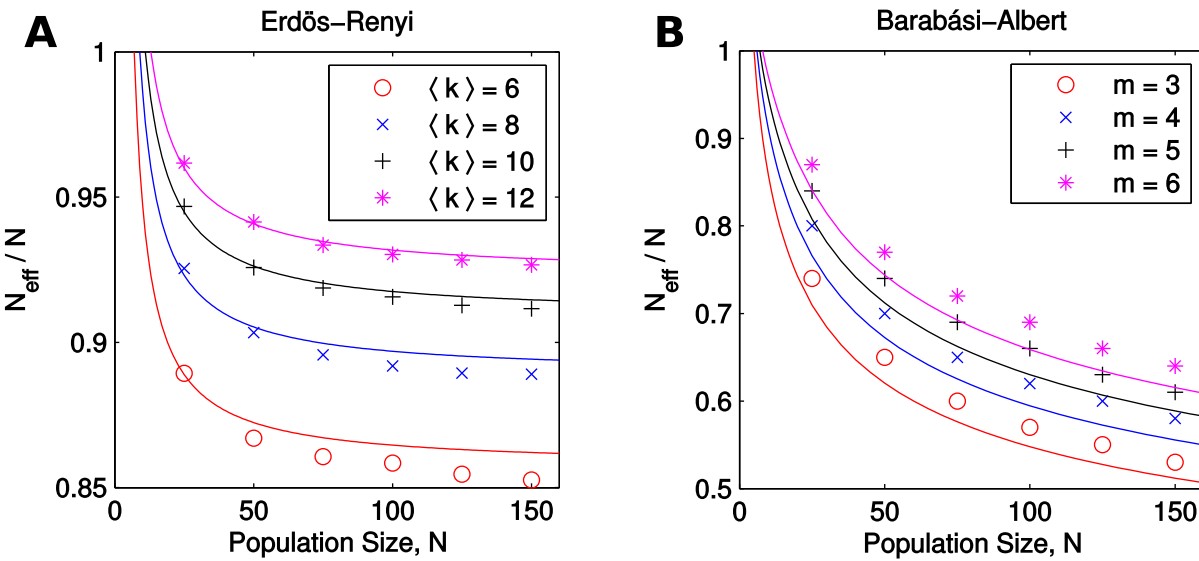

**Fig 6. Random graphs suppress weak selection.** Plot markers show the ratio $N_{eff}/N$, averaged over 1000 trials, plotted against population size $N$. Effective population size, $N_{eff}$, is calculated by numerically solving Eq (5) for each graph and applying Eqs (6) and (9). All random graphs generated have $N_{eff} < N$ and are therefore suppressors of weak selection. Curves of the corresponding colors show the approximation $N_{eff}/N \approx \mu_1^2/\mu_2$ from Eq (24). Overall, we find that larger, sparser, and more heterogeneous graphs have smaller $N_{eff}/N$; these trends are all reflected in the approximation from Eq (24). (**A**) Erdös-Renyi graphs were generated for specific values of the expected degree $\langle k \rangle$ by setting the link probability to $p = \langle k \rangle/(N-1)$. The moments $\mu_1$ and $\mu_2$ were approximated by assuming that the degree of each vertex is independently distributed as $\text{Binom}(N-1, p)$. This leads to $N_{eff}/N \approx (N-1)p/[(N-2)p+1]$. At the minimum population size of $N = \langle k \rangle + 1$, the graph is complete and therefore $N_{eff}/N = 1$. (**B**) Barabási-Albert preferential attachment networks [64] were generated for linking numbers $3 \le m \le 6$, starting from a complete graph of size $m + 2$. The second moment was calculated using the expected degree distribution for finite Barabási-Albert networks obtained by Fotouhi and Rabbat [65]. At the minimum population size of $N = m + 2$, the graph is complete and therefore $N_{eff}/N = 1$.

nor can it detect complex behavior such as multiple switchings between amplification and suppression [27].

## Effective population size

Our analysis motivated a new definition of the effective population size of a graph, $N_{eff} = \sum_{i \in G} \pi_i \tau_i$. This notion of effective population is particular to dB updating, since it was derived from weak-selection fixation probabilities under this update rule. Our definition has a number of interesting connections to other definitions previously proposed for this concept [16, 38–40].

First, as noted above, the effective population size of Antal et al. [16] appears in Eq (24) as an approximation to ours. Whereas we obtain $N_{eff} \approx N\mu_1^2/\mu_2$ using coalescent theory and assuming uniformity of remeeting times, Antal et al. [16] obtain the same expression using diffusion approximation and assuming degree-uncorrelatedness of the graph. That the same expression arises from distinct analytical frameworks and assumptions hints at its naturality.

Second, our definition differs by a simple rescaling from the notion of "fixation effective population size" proposed by Allen, Dieckmann, and Nowak (hereafter, ADN) [39], and elaborated upon by Giaimo et al. [40]:

$$N_{eff}^{ADN} = \frac{N^2}{N-1} \frac{d\rho}{dr}\Big|_{r=1}. \tag{25}$$

Comparing Eqs (25) and (8), we find the relationship

$$N_{eff}^{ADN} = \frac{N(N_{eff} - 2)}{2(N-1)}.$$

For large populations, $N_{\text{eff}}^{\text{ADN}} \approx N_{\text{eff}}/2$. The factor of two appears because the ADN definition uses the Wright-Fisher (discrete generations) model as a baseline, whereas the baseline for our $N_{\text{eff}}$ is the death-Birth process, for which generations are overlapping. Such factors of two commonly appear in translating between discrete- and overlapping-generations models [36, 39, 55].

Third, our proposed definition is closely related to the concept of "inbreeding effective population size", which dates back to Wright [34] and has been elaborated on by many others [35–38]. The inbreeding effective population size is typically defined, for diploid populations, as the size of an idealized population that would have the same level of autozygosity (a locus containing two alleles that are identical by descent) [35, 37]. Although autozygosity as such cannot occur in haploid populations, the remeeting time $\tau_i$ quantifies the closely-related concept of *auto-coalescence*—the time for two hypothetical, independent lineages from $i$ to coalesce. For rare mutation, coalescence time is proportional to the probability of non-identity by descent [58]; thus auto-coalescence can be taken as a proxy for autozygosity in haploid populations. Our $N_{\text{eff}}$ is equal to the size of a well-mixed population that would experience the same degree of auto-coalescence, when averaged over individuals weighted by their reproductive values $\pi_i$. It is therefore reasonable to interpret our $N_{\text{eff}}$ as a haploid analogue of the inbreeding effective population size.

## Transient amplifiers of selection

The most novel of our results is the discovery of the first transient amplifiers of selection for dB updating. Previous investigations [18, 23, 28] had uncovered only suppressors and reducers. Of the transient amplifiers we have found, the strongest is the 2-Fan, $F_{n,2}$, with many blades ($n \to \infty$; Fig 3D). A companion work [46] proves that full (non-transient) amplifiers cannot exist for death-Birth updating.

Transient amplifiers appear to be quite rare for death-Birth updating. None were present within an ensemble of thousands of small graphs analyzed by Hindersin and Traulsen [23]. Similarly, no amplifiers of weak selection for dB were found in our ensembles of Erdös-Renyi and Barabasi-Albert random graphs.

Why should transient amplifiers be so rare? One possible clue comes from the approximation for effective population size in Eq (24). The approximated $N_{\text{eff}}$ is always less than or equal to the actual population size $N$, with equality only for isothermal graphs. Thus any amplifier (transient or not) must be a graph for which the approximation in Eq (24) is inaccurate. Another possible clue is found by combining Eqs (9) and (7) to obtain

$$\frac{N - N_{\text{eff}}}{N^2} = \left(\frac{1}{N} \sum_{i \in G} \pi_i^2 \tau_i\right) - \left(\frac{1}{N} \sum_{i \in G} \pi_i\right)\left(\frac{1}{N} \sum_{i \in G} \pi_i \tau_i\right).$$

The right-hand side can be interpreted as the covariance of $\pi_i$ with $\pi_i \tau_i$, as $i$ runs over vertices of $G$. It follows that $G$ is an amplifier of weak selection if and only if $\pi_i$ and $\pi_i \tau_i$ are negatively correlated on $G$. This requires a very strong negative relationship between weighted degree and remeeting time, which seems unlikely to arise in the usual random graph models. A third clue comes from a companion work [46], which proves a bound on the strength of transient amplifiers for dB. Since transient amplifiers are limited in their possible strength, it is reasonable to suppose they are also limited in number. Each of these clues, however, falls very short of a formal proof.

## Reducers of fixation

Evolutionarily speaking, reducers of fixation maintain the status quo. They protect the resident type from replacement by any mutation, whether beneficial or deleterious. Reducers may have applications in bio-engineering, in situations where it is desirable to inhibit the accumulation of all fitness-affecting mutations. Indeed, it has been argued that the cycle-like structure of epithelial stem cells in mammals [57, 58] may have been evolutionarily designed to limit somatic mutations [30]. The cycle was the first known reducer [23]; others were identified by Hindersin et al. [30]. To these examples we have added two more: the Separated Hubs graph with $n = h + 1$ and the Star of Islands with $m = h$.

Isothermal graphs appear to be obvious candidates for reducers of fixation. This is because, if $G$ is a reducer of fixation, then $\rho_G(r)$ and $\rho_{K_N}(r)$ must coincide to first order in $r$ at $r = 1$, and this latter property holds for all isothermal graphs according to the Weak-Selection Isothermal Theorem for dB. Indeed, all previously-known examples of reducers [23, 30] were isothermal. However, neither the Separated Hubs graph for $n = h + 1$ nor the Star of Islands for $m = h$ are isothermal; thus reducers need not be isothermal. The converse question—whether all isothermal graphs are reducers—remains open. To resolve this question, one would have to either discover or rule out other behaviors for isothermal graphs $G$, such as $\rho_G(r) > \rho_{K_N}(r)$ for all $r$ sufficiently close but not equal to 1. Another open question is whether reducers of fixation exist for Bd updating.

## Limitations

Although we have uncovered an interesting range of behaviors for dB updating on graphs, there are limitations to our approaches. All of our analytical results involve the limit of either weak selection or certain edge weights going to zero. Some of our results combine these limits, meaning that they apply only in rather extreme scenarios, and the results may depend on the limit ordering [59].

We also do not consider the issue of fixation time [33, 41–43, 60–63]. Previous work [42, 43, 60, 61] has uncovered a tradeoff between fixation probability and time: Graphs that amplify selection tend to have larger fixation times than the complete graph, which impedes their ability to accelerate adaptation. A number of our examples involve limits as certain edge weights go to zero. Fixation times diverge to infinity for these examples; therefore they do not hasten the accumulation of beneficial mutations. The search for graphs that (transiently) amplify selection without greatly increasing fixation times is left to future work.

## Conclusion

The identification of amplifiers and suppressors of selection has become a robust field of inquiry [3, 7, 8, 16–29, 40, 42]. Most investigations of this question follow the lead of the initial work [3] in focusing on Birth-death updating. This is an interesting contrast to the study of games on graphs [4, 5, 9–15], which typically considers death-Birth updating—likely because Birth-death updating tends not to support cooperative behaviors [4, 15].

Since the choice of update rule has such marked consequences, a full understanding of evolutionary dynamics in structured populations requires studying a variety of update rules. Indeed, the update rule should properly be considered an aspect of the population structure, equal in importance to the graph itself [11, 15, 28, 29, 52]. If the theory of amplifiers and suppressors is to find application (for example, to microbial populations [8]), it is critical to determine which update rules are plausible for specific organisms. Our work shows that dB

updating exhibits at least some of the interesting phenomena that have been observed for Bd updating, and suggests there is more to be discovered.

## Supporting information

**S1 Appendix. Mathematical derivations and proofs.** This supplement contains a derivation of our weak selection method, Eqs (5)–(9), as well as analysis of our three example graph families.
(PDF)

**S1 Fig. Monte Carlo simulations for weak selection.** Fixation probability approximated from $10^4$ Monte Carlo trials, for $\epsilon = 0.1$, plotted against mutant fitness, $r$ (blue dots). Black lines show the linear approximation to fixation probability as calculated from our weak-selection results; i.e., $\rho \approx 1/N + [(N_{\text{eff}} - 2)/(2N)](r - 1)$ as in Eq (8). As expected, this approximation is accurate for $r \approx 1$. (**A**) The Fan graph, $F_{4,2}$, with $N_{\text{eff}}$ given by Eq (14). (**B**) The Separated Hubs, graph $SH_{3,2,2}$, with $N_{\text{eff}}$ given by Eqs. (40)–(42) of S1 Appendix. (**C**) The Star of Islands graph $SI_{2,3,3}$, with $N_{\text{eff}} \approx 8.89$ as calculated in Mathematica from Eqs (5), (6), and (9).
(EPS)

**S2 Fig. Monte Carlo simulations for nonweak selection with small $\epsilon$.** Fixation probability approximated from $10^4$ Monte Carlo trials, for $\epsilon = 10^{-3}$, plotted against mutant fitness, $r$ (blue dots). Our analytical results for fixation probability in the $\epsilon \to 0$ limit (black curves) show excellent agreement with the simulation results. (**A**) The Fan $F_{4,2}$, with $\epsilon \to 0$ limit given by Eq (15). (**B**) The Separated Hubs graph $SH_{3,2,2}$, with $\epsilon \to 0$ limit given by Eq (17), and (**C**) The Star of Islands graph $SI_{2,3,3}$, with $\epsilon \to 0$ limit given by Eqs (20)–(22).
(EPS)

## Acknowledgments

We are grateful to Josef Tkadlec and Andreas Pavlogiannis for helpful conversations.

## Author Contributions

**Conceptualization:** Benjamin Allen, Christine Sample, Gabor Lippner.

**Data curation:** Christine Sample.

**Formal analysis:** Benjamin Allen, Christine Sample, Robert Jencks, James Withers, Patricia Steinhagen, Lori Brizuela, Darren Parke, Gabor Lippner, Yulia A. Dementieva.

**Funding acquisition:** Benjamin Allen.

**Investigation:** Benjamin Allen, Christine Sample, Robert Jencks, James Withers, Patricia Steinhagen, Lori Brizuela, Joshua Kolodny, Darren Parke, Yulia A. Dementieva.

**Methodology:** Benjamin Allen, Christine Sample, Robert Jencks, Gabor Lippner.

**Project administration:** Christine Sample.

**Software:** Lori Brizuela, Joshua Kolodny, Gabor Lippner.

**Supervision:** Benjamin Allen, Christine Sample, Yulia A. Dementieva.

**Visualization:** Patricia Steinhagen.

**Writing – original draft:** Benjamin Allen, James Withers.

**Writing – review & editing:** Benjamin Allen, Christine Sample.

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
