## [Decision Letter · Decision Letter 0]

20 Aug 2019

Dear Dr Allen,

Thank you very much for submitting your manuscript, 'Transient amplifiers of selection and reducers of fixation for death-Birth updating on graphs', to PLOS Computational Biology. As with all papers submitted to the journal, yours was fully evaluated by the PLOS Computational Biology editorial team, and in this case, by independent peer reviewers. The reviewers appreciated the attention to an important topic but identified some aspects of the manuscript that should be improved.

We would therefore like to ask you to modify the manuscript according to the review recommendations before we can consider your manuscript for acceptance. Your revisions should address the specific points made by each reviewer and we encourage you to respond to particular issues Please note while forming your response, if your article is accepted, you may have the opportunity to make the peer review history publicly available. The record will include editor decision letters (with reviews) and your responses to reviewer comments. If eligible, we will contact you to opt in or out.raised.

- Supporting Information uploaded as separate files, titled 'Dataset', 'Figure', 'Table', 'Text', 'Protocol', 'Audio', or 'Video'.

We hope to receive your revised manuscript within the next 30 days. If you anticipate any delay in its return, we ask that you let us know the expected resubmission date by email at ploscompbiol@plos.org.

Sincerely,

Arne Traulsen

Associate Editor

PLOS Computational Biology

Natalia Komarova

Deputy Editor

PLOS Computational Biology

[LINK]

Reviewer's Responses to Questions

**Comments to the Authors:**

Reviewer #1: A very well written paper that can be accepted as is.

Only one minor issue: There is a missing reference to an equation in the supplemental material (on the line between eqs (12) and (13))

Reviewer #2: This submission is nice, with interesting results that are relatively easy to read, and good tie-ins to existing literature.

My only substantial problem with the submission is in the sections evaluating the methodology for particular graphs, specifically a 'fan.' I see two problems.

First, the analysis investigates a limit of the fixation probability as epsilon approaches 0. But in that limit, the graphs are no longer well-behaved in the sense that their fixation time goes to infinity. In other words, the graph cannot achieve fixation in finite time in the limit that the blades are not connected to the hub. So this limit does not make sense to me, because the graph reduces to independent Moran processes, one on each blade.

Second, there is an error in computing that limit for the fan from Eq. 13. Lim eps -> 0 of Neff (Eq. 13) is mn. Either the equation is misreported or there is a deeper issue to be addressed.

I would also be interested to see a comparison of the theoretical results with simulations that approximate fixation over a large number of trials. For example, in Fig 2 panels C and D, I suggest the authors present simulation results of fixation probability as dots for various values of r and show that Eq 14 passes through them when r is approximately 1. Same thing for Figs 3 and 4. Showing agreement between theory and simulations justifies your methodology and assures the reader that your math is right.

There are a number of additional minor revisions that I include in the attached, including suggested references, interpretations, a request for an additional clarifying figure, etc.

There are unlabelled equations and references in the SI as well.

Reviewer #3: The review is in the attachment.

**Have all data underlying the figures and results presented in the manuscript been provided?**

Reviewer #1: Yes

Reviewer #2: Yes

Reviewer #3: Yes

PLOS authors have the option to publish the peer review history of their article (what does this mean?). If published, this will include your full peer review and any attached files.

Reviewer #1: No

Reviewer #2: Yes: Travis Monk

Reviewer #3: Yes: Marius Moeller

---

## [Decision Letter · Decision Letter 1]

30 Oct 2019

Dear Dr Allen,

We are pleased to inform you that your manuscript 'Transient amplifiers of selection and reducers of fixation for death-Birth updating on graphs' has been provisionally accepted for publication in PLOS Computational Biology.

In the meantime, please log into Editorial Manager at https://www.editorialmanager.com/pcompbiol/, click the "Update My Information" link at the top of the page, and update your user information to ensure an efficient production and billing process.

One of the goals of PLOS is to make science accessible to educators and the public. PLOS staff issue occasional press releases and make early versions of PLOS Computational Biology articles available to science writers and journalists. PLOS staff also collaborate with Communication and Public Information Offices and would be happy to work with the relevant people at your institution or funding agency. If your institution or funding agency is interested in promoting your findings, please ask them to coordinate their releases with PLOS (contact ploscompbiol@plos.org).

Thank you again for supporting Open Access publishing. We look forward to publishing your paper in PLOS Computational Biology.

Sincerely,

Arne Traulsen

Associate Editor

PLOS Computational Biology

Natalia Komarova

Deputy Editor

PLOS Computational Biology

Reviewer's Responses to Questions

**Comments to the Authors:**

Reviewer #3: I thank the authors for taking my comments seriously and making the needed changes in the supplementary. I especially liked the idea of using the perturbation of an isothermal graph to find an unweighted, undirected amplifier.

Looking through the supplement, I couldn't find any further mistakes. As such, I'm very happy with the manuscript as-is.

**Have all data underlying the figures and results presented in the manuscript been provided?**

Reviewer #3: None

PLOS authors have the option to publish the peer review history of their article (what does this mean?). If published, this will include your full peer review and any attached files.

Reviewer #3: Yes: Marius Moeller

---

## [Editor Report · Acceptance letter]

15 Nov 2019

PCOMPBIOL-D-19-00886R1 

Transient amplifiers of selection and reducers of fixation for death-Birth updating on graphs

Dear Dr Allen,

I am pleased to inform you that your manuscript has been formally accepted for publication in PLOS Computational Biology. Your manuscript is now with our production department and you will be notified of the publication date in due course.

With kind regards,

Matt Lyles
